# Longitudinal Changes in IgG-Type SARS-CoV-2 Antibody Titers after COVID-19 Vaccination and a Prominent Increase in Antibody Titers When Infected after Vaccination

**DOI:** 10.3390/vaccines11040860

**Published:** 2023-04-17

**Authors:** Hiroshi Kusunoki, Michiko Ohkusa, Rie Iida, Ayumi Saito, Mikio Kawahara, Kazumi Ekawa, Nozomi Kato, Keita Yamasaki, Masaharu Motone, Hideo Shimizu

**Affiliations:** 1Department of Internal Medicine, Osaka Dental University, 8-1 Kuzuhahanazonocho, Hirakata 573-1121, Japan; 2Department of Laboratory Medicine, Osaka Dental University Hospital, Osaka 565-0871, Japan; 3Department of Environmental and Preventive Medicine, Hyogo Medical University, Nishinomiya 663-8501, Japan; 4Department of Health and Sports Sciences, Graduate School of Medicine, Osaka University, Suita 565-0871, Japan; 5Faculty of Health Sciences, Osaka Dental University, Hirakata 573-1144, Japan

**Keywords:** COVID-19, SARS-CoV-2, vaccines, breakthrough infection, SARS-CoV-2 antibody

## Abstract

Objective: Severe acute respiratory syndrome coronavirus 2 (SARS-CoV-2) antibody titers level and duration of elevated levels are considered important indicators for confirming the efficacy of coronavirus disease 2019 (COVID-19) vaccines. The objective of this study was to demonstrate the changes in antibody titers after the second and third doses of the COVID-19 vaccine, and to determine the antibody titers in cases of spontaneous infection with SARS-CoV-2 after vaccination. Materials and Methods: From June 2021 to February 2023, IgG-type SARS-CoV-2 antibody titers were measured in 127 participants, including 74 outpatients and 53 members of staff, at the Osaka Dental University Hospital (64 males and 63 females, mean age 52.3 ± 19.0 years). Results: Consistent with previous reports, the SARS-CoV-2 antibody titer decreased with time, not only after the second dose but also after the third dose of the vaccine if there was no spontaneous COVID-19 infection. We also confirmed that the third booster vaccination was effective in increasing the antibody titer. A total of 21 cases of natural infections were observed after administering two or more doses of the vaccine. Thirteen of these patients had post-infection antibody titers exceeding 40,000 AU/mL, and some cases continued to maintain antibody titers in the tens of thousands of AU/mL even after more than 6 months had passed since infection. Conclusions: The rise in and duration of antibody titers against SARS-CoV-2 are considered important indicators for confirming the efficacy of novel COVID-19 vaccines. A longitudinal follow-up of antibody titers after vaccination in larger studies is warranted.

## 1. Introduction

Severe acute respiratory syndrome coronavirus 2 (SARS-CoV-2) antibody titers and the duration of elevated levels are considered important indicators of the efficacy of novel coronavirus vaccines. In Japan, with the spread of coronavirus disease 2019 (COVID-19), many medical institutions offer the SARS-CoV-2 antibody titer test as an uninsured, self-paid test. However, there are few reports on antibody titer testing from such medical institutions. In Japan, SARS-CoV-2 mRNA vaccination began in February 2021, and more than 70% of the population had been vaccinated at least twice by the end of 2021. In Japan, the two most commonly used mRNA vaccines for COVID-19 are the BNT162b2 mRNA vaccine (BioNTech and Pfizer) and the mRNA-1273 vaccine (Moderna and Takeda). It is generally known that antibody titers after the SARS-CoV-2 vaccination decay over several months. In Japan, COVID-19 has spread rapidly due to the sixth wave of infection since the beginning of 2022, which was mainly caused by the Omicron strain.

Since the number of patients infected after the sixth wave was overwhelmingly larger than the cumulative number of those infected before the sixth wave, it can be assumed that breakthrough infection with COVID-19 after two or more vaccinations is very common. However, few reports have followed SARS-CoV-2 antibody titers over time after spontaneous infection following vaccination.

The Department of Internal Medicine, Osaka Dental University Hospital, has been measuring IgG-type SARS-CoV-2 antibody titers using the Architect SARS-CoV-2 IgG II Quant (Abbott Laboratory) in outpatients and medical staff since June 2021.

In this article, we will show the relationship between the number of days after the second and third booster doses and antibody titers, as well as the evolution of antibody titers in cases of COVID-19 infection after more than two doses of the vaccine.

## 2. Materials and Methods

This was a single-center retrospective study. We conducted a longitudinal observational study involving outpatients and medical staff at the Osaka Dental University Hospital. The study protocols were approved by the ethics committee of Osaka Dental University Hospital (2022-10). Written informed consent was obtained from all participants. From June 2021 to February 2023, IgG-type SARS-CoV-2 antibody titers were measured in 127 participants, including 74 outpatients and 53 staff members at the Osaka Dental University Hospital (64 males and 63 females, mean age 52.3 ± 19.0 years) (Table 1). The titers of SARS-CoV-2 anti-receptor-binding domain IgG antibodies were measured in serum samples.

First, the correlation between antibody titer, subject age, and the number of days after the second dose of the vaccine was examined in 56 subjects (30 males and 26 females) with no history of COVID-19 infection, and the date of vaccination was clear. The same examination was also conducted on 23 subjects (12 males and 11 females) with no history of COVID-19 infection after the third dose of the vaccine, and the date of vaccination was clear (Figure 1). The correlation between pre- and post-vaccination antibody titers was examined in 12 subjects (six males and six females) who had antibody titers measured before and after the third dose of the vaccination.

The post-infection antibody titers of subjects infected with COVID-19 after two or more doses of the COVID-19 vaccine were also examined.

## 3. Serology Assays

We used the Abbott Architect SARS-CoV-2 IgG II Quant (Abbott Laboratories, Chicago, IL, USA) chemiluminescent microparticle immunoassay to detect IgG antibodies to the receptor-binding domain of the S1 subunit of the SARS-CoV-2 spike protein, according to the manufacturer’s instructions. The reportable measurement range of the assay is up to 40,000 AU/mL. IgG antibody titers >50 AU/mL (the cut-off set by the manufacturer) were considered indicative of seropositivity.

## 4. Statistical Analysis

The results are expressed as the mean ± standard deviation (SD). Pearson’s product-moment correlation coefficient was used to assess the associations between SARS-CoV-2 anti-receptor binding domain IgG antibody titers, age, and number of days after vaccination. For data analysis, the JMP 13.1 software was used for data analysis. Statistical significance was set at *p* < 0.05.

## 5. Results

Of the 127 enrolled participants (64 males and 63 females) whose SARS-CoV-2 antibody titers were measured at the Osaka Dental University Hospital, 120 participants (62 males and 58 females) were vaccinated up to the second dose, 73 participants (36 males and 37 females) up to the third dose, and six were never vaccinated. One participant complained of feeling sick after the first dose of the vaccine and did not receive the second dose (Table 1, Figure 1).

Regarding the type of vaccine, many had received BNT162b2 up to the second and third doses, and a few had received mRNA-1273, but many were unsure of the vaccine they had received (Table 1).

Antibody titers were measured in 56 subjects (30 males and 26 females) after the second dose of the vaccine. These subjects had no history of COVID-19 infection at the time of the antibody titer measurement. A tendency for antibody titers to decrease was observed (Figure 2). The median age was 59.5 years, the mean antibody titer was 4807.7 ± 6472.6 AU/mL, and the median antibody titer was 2052.0 U/mL (Table 2). The antibody titer was negatively correlated with the number of days after the second vaccine dose (*p* < 0.001). It has previously been reported that antibody titers after vaccination tend to decrease with increasing age [1,2,3,4]. A mild correlation between age and antibody titer was observed in our cohort of subjects, but the association was not statistically significant (*p* = 0.066).

Figure 3 shows the longitudinal changes in antibody levels in eight patients (four males and four females) with several antibody titer follow-ups after the second dose of the vaccine. These eight patients had no history of COVID-19 infection during that time. The first antibody titer was measured within 1 month of the second dose of the vaccination, but the titers varied greatly among individuals, ranging from several thousand to several tens of thousands AU/mL. The titers decreased by half to one-third in 3 months, and by 6 months, they had decreased to a few hundred AU/mL.

For these eight cases, we compared the maximum antibody titer and the minimum antibody titer. The antibody titer decreased remarkably before the third dose of the vaccine. When the difference between the maximum and minimum titers was divided by the number of days between the two samplings, it was found that the higher the maximum value, the higher the rate of decrease in the antibody titer per day (Table 3).

Antibody titers were measured in 23 subjects (12 males and 11 females) with no history of COVID-19 infection at the time of antibody titer measurement after the third booster dose of the vaccine (Figure 4A). The average antibody titer after the third dose of the vaccine was 12,617.5 ± 7256.2 AU/mL, and the median antibody titer was 14,416.2 AU/mL, which was certainly boosted from that after the second dose of the vaccine. The number of days between the third dose of the vaccine and the antibody titer measurement date also showed a negative correlation with the antibody titer, as did the titer after the second dose of the vaccine (*p* = 0.039). As in the case after the second dose of the vaccine, there may be a slight correlation with age, but this was not significant in our cohort of subjects (*p* = 0.689).

Figure 4B shows the scattered plotting of the antibody titers before and after the third booster dose of the vaccine for the 12 subjects (6 males and 6 females) from whom serum samples could be obtained before and after the third booster dose of the vaccine. This number of subjects did not result in a significant correlation (*p* = 0.074), but the correlation coefficient was above 0.5, indicating a significant correlation. It was suggested that a positive correlation exists between antibody titers before and after the third booster dose. We also found that the average antibody titer before the third booster dose of the vaccine was in the range of several hundred AU/mL, but after the third booster dose, many of the subjects had titers above 10,000 AU/mL.

Next, the cases of patients with boosted antibody titers after two doses of the vaccine who developed a COVID-19 infection are shown (Figure 5A). The antibody titer of a 41 year-old male before the first dose of the vaccine in July 2021 was negative, and he had no episode of suspected COVID-19 infection up until that time. After the second dose of vaccine, his antibody titer jumped to nearly 27,000 AU/mL and then decreased to about 4000 AU/mL in approximately six months. When the antibody titers were measured in April 2022, he was considering the third dose of vaccine, but his titers increased to over 40,000 AU/mL, which was well above the limit of measurement. In other words, it is thought that this participant was spontaneously infected with COVID-19 somewhere from February to March 2022, but PCR testing was not conducted, and the participant was not counted as a COVID-19-infected patient. Even nine months after the suspected COVID-19 infection, the antibody titer remained above 40,000 AU/mL.

There have been other cases of spontaneous COVID-19 infection after a second dose of the vaccine. A 65 year-old male received a second dose of vaccine in April 2021 (Figure 5B). Thereafter, his antibody titer declined, temporarily falling to 600 AU/mL. However, he was spontaneously infected with COVID-19 in February 2022, and his antibody titer increased to over 40,000 AU/mL. When we followed up on the antibody titer in January 2023, nearly one year after the COVID-19 infection, the antibody titer was still high (>10,000 AU/mL).

A 39 year-old female had completed her second dose of vaccine by May 2021, and her antibody titer had decreased to approximately 300 AU/mL before the third dose of vaccine in February 2022 (Figure 5C). The third dose of the vaccine in February boosted her antibody titer to >20,000 AU/mL. Although her antibody titer was boosted by the third dose of the vaccine, she spontaneously developed COVID-19 in May. After the COVID-19 infection, her titer increased to over 40,000 AU/mL. Six months after the COVID-19 infection, the antibody titer remained high, at 37,000 AU/mL.

There was a case of spontaneous COVID-19 infection after a third dose of the vaccine. A 60 year-old male had completed his second dose of vaccine by June 2021 and received a third dose of vaccine in February 2022 (Figure 5D). After the third dose of vaccine, his antibody titer rose to more than 10,000 AU/mL, and then decreased to about 4800 AU/mL in approximately two months. When the antibody titers were measured in September 2022, his titers increased to over 40,000 AU/mL. Like the case of Figure 5A, it is thought that this participant was spontaneously infected with COVID-19 somewhere from May to September 2022, but PCR testing was not conducted, and the participant was not counted as a COVID-19-infected patient. Even six months after the suspected COVID-19 infection, the antibody titer remained above 30,000 AU/mL.

There were 17 cases of spontaneous COVID-19 infections after receiving two or more doses of vaccines (Table 4). All patients were infected with COVID-19 after two or three doses of the vaccine, and all of them were either asymptomatic or showed very mild symptoms. Some of them were not counted as infected people because PCR tests were not performed. Nine of them had antibody titers of more than 40,000 AU/mL after spontaneous infection, and two of them maintained antibody titers in the 10,000 AU/mL range even after more than six months had passed since infection.

## 6. Discussion

The COVID-19 vaccine has been available in Japan since February 2021 for healthcare workers and since April 2021 for elderly citizens. As of 10 December 2021, 77.3% of the total population had completed two vaccination doses. The third dose of vaccinations also began in December 2021 for healthcare workers, and in early 2022, vaccinations began in earnest for the general population, starting with the elderly.

Despite the expansion of COVID-19 vaccination, the Omicron strain has spread tremendously since the beginning of 2022. As of February 2023, the cumulative number of people infected with COVID-19 in Japan exceeded 32 million, most of whom were infected after 2022, when the Omicron strain became the dominant strain.

The elevation and duration of the SARS-CoV-2 neutralizing antibody titer are considered important indicators to confirm the efficacy of the COVID-19 vaccine [5]. In addition, the anti-receptor-binding domain (RBD) antibody, an antibody against the SARS-CoV-2 spike protein, is widely used in Japan because it correlates with the SARS-CoV-2 neutralizing antibody titer [6,7] and can be measured at general medical facilities.

Antibodies against the SARS-CoV-2 spike protein after COVID-19 vaccination have been reported to decay over several months [8,9]. In this study, consistent with previous reports, we confirmed that the SARS-CoV-2 antibody titer decreases with time after the second dose of the vaccine if there is no spontaneous COVID-19 infection.

Anti-receptor-binding domain (RBD) antibody titers are known to increase markedly after the booster dose (third dose) of the vaccine [10]. In this study, consistent with previous reports, we have confirmed that the third booster vaccination is effective in increasing the antibody titer, but the boosted antibody titer after the third vaccination also decreased over time. This tendency is consistent with previous reports [11]. Antibody titers before the third dose of the vaccine tended to correlate with the boosted antibody titer after the third booster dose. This result is also consistent with previous reports [12].

It has long been known that people who were previously infected with COVID-19 have markedly increased antibody titers after vaccination compared to uninfected people [13,14,15]. It has been suggested that the vaccination of previously infected patients with COVID-19 is questionable [16]. Breakthrough infections of COVID-19 after vaccination have become a problem. In Japan, breakthrough infections frequently occur after two or more vaccine doses. It has been reported that breakthrough infection after vaccination markedly increases antibody titers [17]. The medical records of a general internal medicine clinic in Wakayama, Japan, indicate that anti-receptor-binding domain (RBD) antibody titers are markedly elevated in breakthrough infections with COVID-19 after vaccination, as was found in this study [18]. However, there are currently few reports of the long-term follow-up of antibody titers increased by breakthrough infections.

In this study, we observed that when patients spontaneously became infected with SARS-CoV-2 after the second dose of vaccine, antibody titers increased to more than 40,000 AU/mL and remained high for several months. Although antibody titers decay after vaccination, it has been reported that not only memory B cells but also cellular immunity, such as memory T cells, persist for six months after vaccination [19]. The decrease in antibody titer may not directly lead to a decrease in infection protection, but the antibody titer may be immediately boosted by spontaneous infection if there is a history of vaccination in the past due to immune memory.

SARS-CoV-2 antibody titers vary greatly from person to person and increase considerably after spontaneous infection. In Japan, it is thought that many people have antibody titers of tens of thousands of AU/mL after spontaneous COVID-19 infection (often very mild and asymptomatic) following vaccination.

In Japan, a large number of patients were infected with the Omicron strain in 2022 after receiving two doses of the COVID-19 vaccine in 2021. As observed in this study, such cases are likely to maintain high antibody titers for more than six months after infection; however, in reality, even those with sufficiently high antibody titers are likely to have received a third or subsequent additional vaccination in a large number of cases. It is questionable whether individuals who were infected with COVID-19, and whose antibody titers rose above the measurable limit and remained high for more than six months after infection, as was the case in the present study, should be uniformly vaccinated.

“Hybrid immunity” of vaccination plus spontaneous infection brought about by breakthrough infections after vaccination is gaining worldwide attention as COVID-19 vaccination becomes more widespread [20]. Hybrid immunity offers protection against COVID-19 infection and is reported to persist for a relatively long time (between 6 and 8 months) [21,22]. A systematic review of recent studies has also shown that individuals with hybrid immunity are more protected against the Omicron variant than those with only a history of infection, suggesting that individuals with hybrid immunity may not require a booster dose immediately [23]. It is expected that high antibody titers will be maintained over a long period of time in those with such “hybrid immunity”.

Although this data is based on a small number of studies at a single institution using a measurement system that has already been commercialized and is widely used, we believe that the changes in antibody titers over time are significant indicators of COVID-19 vaccinations, especially in cases of breakthrough infection. In addition, some of these post-vaccination breakthrough infection cases were PCR-untested and had not been definitively diagnosed with COVID-19. This suggests that there may be hidden cases with very high antibody titers due to hybrid immunity among those who are asymptomatic, very mildly infected with COVID-19, unaware, and undiagnosed. Although the existence of such cases is within the range of expectations, it is significant that we were able to use a real case in our demonstration.

This study has several limitations. First, the size of the sample is small. Second, there is no information on SARS-CoV-2 variants. Third, the natural immunity to SARS-CoV-2 after vaccination has not yet been studied. In addition to the induction of antibodies, the effect of natural immunity is considered significant in terms of hybrid immunity. As for cellular immunity after COVID-19 infection, the effects of CD4+ T cells and CD8+ T cells have been shown to be maintained for more than eight months [24,25]. The analysis of peripheral blood mononuclear cells (PBMCs) from COVID-19 patients is known to show impaired expression of inflammatory cytokines [26]. Although PBMCs were not analyzed in this study, future analysis of PBMCs from post-vaccination breakthrough infection cases may shed more light on the importance of natural immunity in “hybrid immunity” in COVID-19.

## 7. Conclusions

In conclusion, the increase in antibody titers against SARS-CoV-2 and the duration for which it remains high are considered important indicators of the efficacy of novel coronavirus vaccines. It is desirable to measure antibody titers and prioritize those with low antibody titers for booster vaccination, rather than blindly recommending booster vaccination to the entire population. As this was a small, single-center retrospective study, larger-scale, longitudinal studies of antibody titers are warranted.

## Figures and Tables

**Figure 1 vaccines-11-00860-f001:**
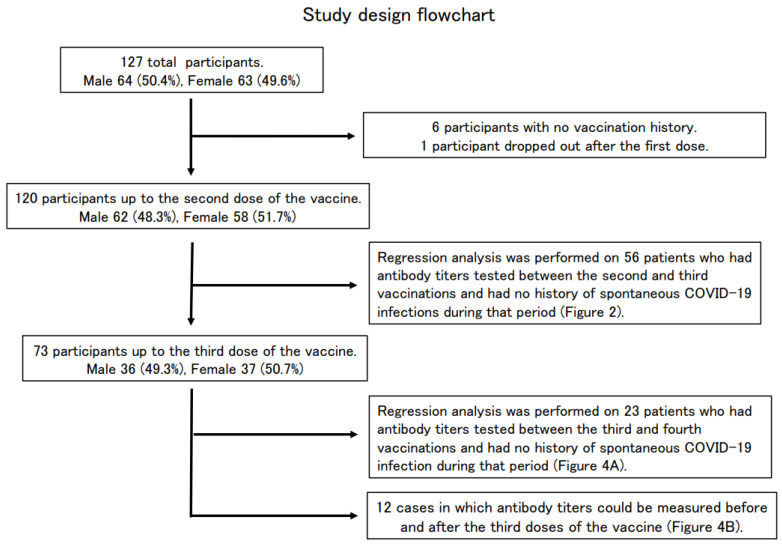
Study design flowchart.

**Figure 2 vaccines-11-00860-f002:**
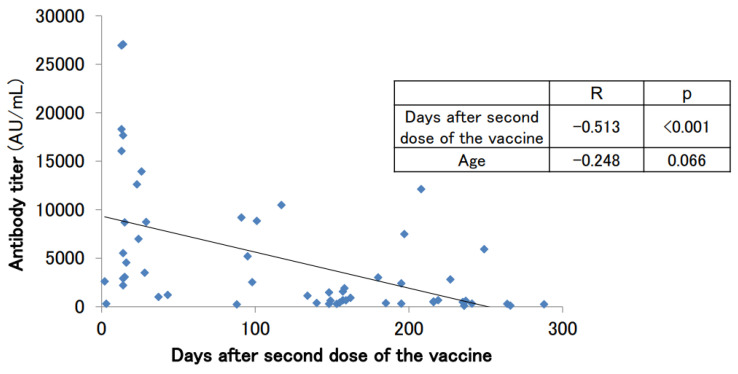
SARS-CoV-2 antibody titers in participants with no history of COVID-19 infection who had antibodies measured after the second dose of the vaccine.

**Figure 3 vaccines-11-00860-f003:**
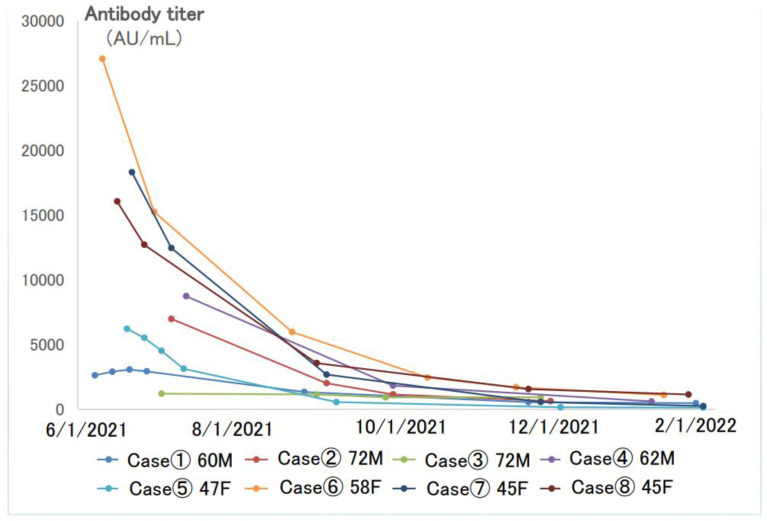
Longitudinal changes of SARS-CoV-2 antibodies in eight cases with multiple follow-ups after the second dose of the vaccine.

**Figure 4 vaccines-11-00860-f004:**
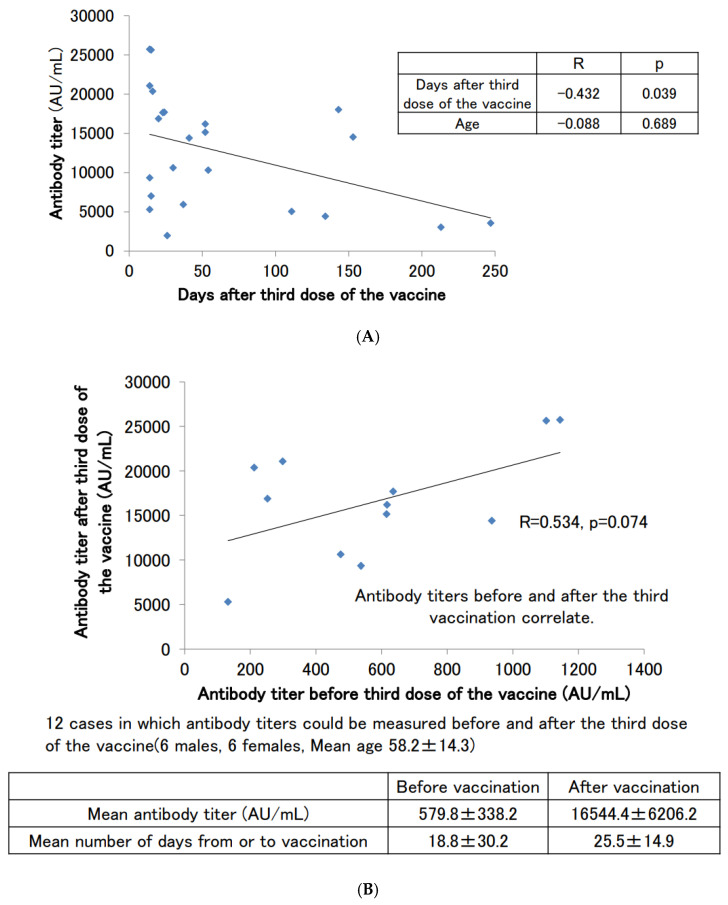
(**A**) SARS-CoV-2 antibody titers in participants with no history of COVID-19 infection who had antibodies measured after the third dose of the vaccine. (**B**) Correlation of SARS-CoV-2 antibody titers before and after the third dose of the vaccine.

**Figure 5 vaccines-11-00860-f005:**
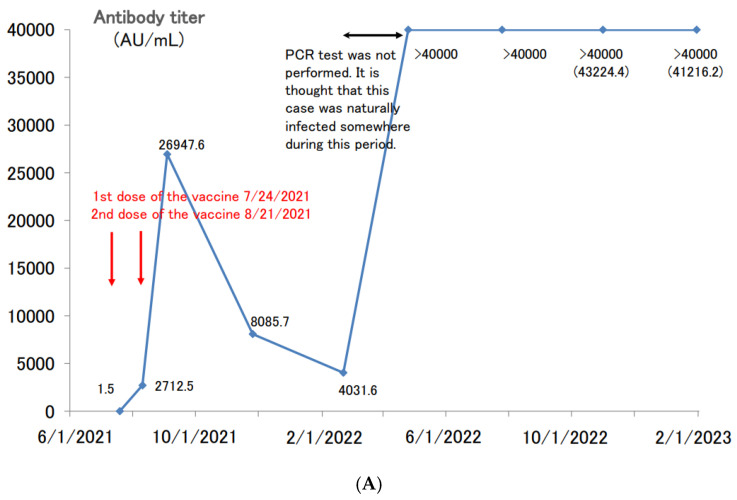
(**A**). Longitudinal change of the SARS-CoV-2 antibody titer of a 41 year-old male. (**B**) Longitudinal change of the SARS-CoV-2 antibody titer of a 65 year-old male. (**C**) Longitudinal change of the SARS-CoV-2 antibody titer of a 39 year-old female. (**D**) Longitudinal change of the SARS-CoV-2 antibody titer of a 60 year-old male.

**Table 1 vaccines-11-00860-t001:** Participant characteristics up to the second and third vaccine doses.

	Total Participants	Participants up to the Second Dose	Participants up to the Third Dose
**Total**	127	120	73
**Male, *n* (%)**	64 (50.4)	62 (48.3)	36 (49.3)
**Female, *n* (%)**	63 (49.6)	58 (51.7)	37 (50.7)
**Mean age**	52.3 ± 19.0	52.7 ± 19.0	54.6 ± 17.8
**Median age**	57	57.5	59
**Type of vaccine**	**BNT162b2 (BioNTech and Pfizer), *n* (%)**	64 (53.3)	56 (76.7)
	**mRNA-1273 (Moderna and Takeda), *n* (%)**	12 (10.0)	8 (11.0)
	**Unknown, *n* (%)**	44 (36.7)	9 (12.3)
**No vaccination history, *n* (%)**	6 (4.7)		
**Participants up to the first dose**	1 (0.8)		

**Table 2 vaccines-11-00860-t002:** Analyzed participants up to the second and third vaccine doses.

		Analyzed Participants up to the Second Dose	Analyzed Participants up to the Third Dose
**Total**		56	23
**Male, *n* (%)**		30 (53.6)	12 (52.2)
**Female, *n* (%)**		26 (46.4)	11 (47.8)
**Mean age**		55.6 ± 16.5	59.3 ± 14.8
**Median age**		59.5	60.0
**Type of vaccine**	**BNT162b2 (BioNTech and Pfizer), *n* (%)**	34 (60.7)	16 (69.6)
	**mRNA-1273 (Moderna and Takeda), *n* (%)**	6 (10.7)	4 (17.4)
	**Unknown, *n* (%)**	16 (28.6)	3 (13.0)
**Mean antibody titer (U/mL) (Mean ± standard deviation)**		4807.7 ± 6472.6	12,617.5 ± 7256.2
**Median antibody titer (U/mL)**		2052.0	14,416.2
**Mean number of days between the vaccination and the measurement day** **(Mean ± standard deviation)**		122.0 ± 89.3	63.6 ± 68.6

**Table 3 vaccines-11-00860-t003:** Eight cases with multiple follow-ups after the second dose of vaccine.

Case	Days from the 2nd Dose of the Vaccine to the Maximum Antibody Titer	Maximum Antibody Titer (AU/mL)	Minimum Antibody Titer (AU/mL)	Follow-up (Number of Days)	Rate of Decrease in Antibody Titer (AU/mL/Day)
Case①60M	21	3073.8	475.5	230	11.3
Case②72M	24	6988.8	635.4	154	41.3
Case③83M	65	1211.7	936.1	154	1.8
Case④62M	29	8745.5	616.8	189	43.0
Case⑤47F	7	6221.9	132.2	234	26.0
Case⑥58F	14	27,079.9	1101.9	228	113.9
Case⑦45F	13	18,316.8	252.3	166	108.8
Case⑧45F	13	16,064.3	1144	232	64.3

**Table 4 vaccines-11-00860-t004:** Cases of spontaneous COVID-19 infection after two or more doses of vaccine.

Case	1st Vaccination Date	2nd Vaccination Date	3rd Vaccination Date	Date of Infection	Post-Infection Antibody Test Date①	Post-Infection Antibody Titer① (AU/mL)	Post-Infection Antibody Test Date②	Post-Infection Antibody Titer② (AU/mL)	Post-Infection Antibody Test Date③	Post-Infection Antibody Titer③ (AU/mL)
**36M**	4/30/2021	5/21/2021		1/17/2022 ※	2/7/2022	>40,000.0	8/16/2022	31,409.9	11/15/2022	22,684.0
**65M**	4/27/2021	5/18/2021	2/7/2022	4/2/2022	4/20/2022	>40,000.0	10/4/2022	15,769.9	12/27/2022	12,379.0
**45M**	5/6/2021	5/27/2021		8/3/2022	9/20/2022	>40,000.0	12/23/2022	19,533.0		
**68M**	July 2021	Aug 2021	April 2022	Aug 2022 #	11/1/2022	>40,000.0	12/20/2022	39,247.6		
**63M**	4/28/2021	5/19/2021	2/4/2022	8/20/2022	11/4/2022	>40,000.0				
**44M**	7/24/2021	8/21/2021	7/16/2022	8/9/2022	11/10/2022	17,451.4				
**27F**	4/30/2021	5/21/2021		1/22/2022	2/7/2022	>40,000.0				
**23F**	5/26/2021	6/18/2021	3/11/2022	Aug 2022	11/28/2022	>40,000.0				
**24F**	April 2021	5/14/2021	1/15/2022	Aug 2022	11/28/2022	36,446.4				
**23F**	7/24/2021	8/21/2021	4/22/2022	8/12/2022	12/1/2022	36,292.0				
**35F**	April 2021	May 2021		1/10/2022	1/24/2022	18,907.4				
**28F**	8/27/2021	9/17/2021	4/9/2022	12/14/2022	1/6/2023	45,218.1				
**25F**	April 2021	Aug 2021	May 2022	Aug 2022	11/29/2022	37,648.8				
**23F**	7/24/2021	8/21/2021	4/22/2022	8/12/2022	12/1/2022	27,238.4				
**23F**	July 2021	8/13/2021	3/19/2022	Sep 2022	12/2/2022	>80,000.0				
**63F**	5/14/2021	6/4/2021	3/14/2022	11/9/2022	12/5/2022	31,879.8				
**40F**	3/26/2021	4/16/2021	12/17/2021	1/1/2023	2/2/2023	27,849.0				

※ No fever, only mild cold symptoms; PCR test was not conducted. # No symptoms due to concentrated contact only; PCR test was not conducted.

## Data Availability

Data cannot be released because it contains personal patient information. Researchers who meet the criteria for access to confidential data may obtain the data via the Clinical Trials Committee. The name of the institution limiting this data is the IRB of Osaka Dental University. If you would like to access the data, please contact us at this telephone number or URL. +81-72-864-3111, https://www.osaka-dent.ac.jp/access.html.

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
