# Peer review of "Longitudinal Changes in IgG-Type SARS-CoV-2 Antibody Titers after COVID-19 Vaccination and a Prominent Increase in Antibody Titers When Infected after Vaccination"

_vaccines, 2023, doi:10.3390/vaccines11040860_

Round 1

Reviewer 1 Report

Dear Authors,

Congratulation to the author. The result proves that breakthrough infection after vaccination increases antibody titers in figures 4A,4B, and 4C. If the study was recruited more volunteers, the data would be interesting.  In addition, the authors would collect whole blood to study cellular immunity. Do you have any data about PBMC in this study?The paper can publish to encourage people to boost vaccine annually to raise an antibody for protection SARS-Cov-2 virus.

Thank you

Cheers,

Thang Truong PhD.

Author Response

Reviewer 1

Congratulation to the author. The result proves that breakthrough infection after vaccination increases antibody titers in figures 4A,4B, and 4C. If the study was recruited more volunteers, the data would be interesting.  In addition, the authors would collect whole blood to study cellular immunity. Do you have any data about PBMC in this study? The paper can publish to encourage people to boost vaccine annually to raise an antibody for protection SARS-Cov-2 virus.

  1. Thank you for your very positive observations. With reference to your comments, I have tried to touch on natural immunity, including cellular immunity, in the “Limitations” section. I have also mentioned the significance of PBMC (Peripheral Blood Mononuclear Cells) analysis, although no data in this regard has been garnered by the study.

L190-L195

“This study has several limitations. First, the size of the sample is small. Second, there is no information on SARS-CoV-2 variants. Third, the natural immunity to SARS-CoV-2 after vaccination has not yet been studied. In addition to the induction of antibodies, the effect of natural immunity is considered significant regarding hybrid immunity. As for cellular immunity after COVID-19 infection, the effects of CD4+ T cells and CD8+ T cells have been shown to be maintained for more than eight months [24,25]. The analysis of peripheral blood mononuclear cells (PBMCs) of COVID-19 patients is known to show impaired expression of inflammatory cytokines [26]. Although PBMCs were not analyzed in this study, future analysis of PBMCs from post-vaccination breakthrough infection cases may shed more light on the importance of natural immunity in "hybrid immunity" in COVID-19.”

Reviewer 2 Report

Dear Author,

This is a well-written scientific MS and it addresses a topic of high importance!

This is a longitudinal observational study including both outpatients and medical staff. The author analyzed the alteration in SARS-CoV-2 antibody levels following the second and third doses of SARS-CoV-2 vaccine and demonstrated the antibody levels in symptomatic COVID-19 that occurred after immunization.

Below, please find minor suggestions:

ABSTRACT:

Line: 20-22 From June 2021 to February 2023, IgG-type SARS-CoV-2 antibody titers were measured in 127 participants, including 74 outpatients and 53 members of staffs, at the Osaka Dental University Hospital (65 males and 62 females, mean age 52.3 ± 19.1 years).

Line: 23-24 Consistent with previous reports, the SARS-CoV-2 antibody titer decreased with time, not only after the second dose, but also after the third dose of the vaccine if there is was no spontaneous COVID-19 infection.

KEYWORDS COVID-19; SARS-CoV-2; vaccines; breakthrough infection; SARS-CoV-2 antibody (The title words should not be repeated in Keywords).

METHODS

Please add tables with “Clinical characteristics of study subjects

DISCUSSION

Please, consider to discuss the results from the unpublished study and use as reference:

Trends in Antibody Titers after SARS-CoV-2 Vaccination - Insights from a Self-Paid Tests at a General Internal Medicine Clinic

Hiroshi Kusunoki, Kazumi Ekawa,  Masakazu Ekawa, Nozomi Kato M, Keita Yamasaki, Masaharu Motone, Hideo Shimizu.

Keywords: COVID-19; SARS-CoV-2; vaccines; breakthrough infection; SARS-CoV-2 antibody

Line: 185-187 In addition, the anti-receptor-binding domain (RBD) antibody, an antibody against SARS-CoV-2 186 spike protein, is widely used in Japan because it correlates with the SARS-CoV-2 neutralizing antibody titer (6,7) and can be measured at general medical facilities.

Line: 193-194 Anti-receptor-binding domain (RBD) antibody titers are known to increase markedly after the booster dose 193 (third dose) of the vaccine (10).

Line: 217-2020 SARS-CoV-2 antibody titers vary greatly from person to person and increase prominently considerably after spontaneous infection. In Japan, it is thought that many people have antibody titers of tens of thousands of AU/mL after spontaneous COVID-19 infection (often very mild and asymptomatic) following vaccination.

Line: 221-222 In Japan, a large number of patients are were infected with the Omicron strain in 2022 after receiving two doses of the COVID-19 vaccine in 2021.

Line: 222-225 As observed in this study, such cases are likely to maintain high antibody titers for more than six months after infection; however, in reality, even those with sufficiently high antibody titers are likely to have received a third or subsequent additional vaccination in a very large number of cases.

Please, highlight the strengths and weaknesses (or limitations) of your study.

Author Response

Reviewer 2

Dear Author,

This is a well-written scientific MS and it addresses a topic of high importance! This is a longitudinal observational study including both outpatients and medical staff. The author analyzed the alteration in SARS-CoV-2 antibody levels following the second and third doses of SARS-CoV-2 vaccine and demonstrated the antibody levels in symptomatic COVID-19 that occurred after immunization.

Below, please find minor suggestions:

  1. Thank you for your favorable comments.

METHODS

Please add tables with “Clinical characteristics of study subjects”

  1. The clinical characteristics of the subjects are summarized in Table 1 and Table 2.

DISCUSSION

Please, consider to discuss the results from the unpublished study and use as reference:

Trends in Antibody Titers after SARS-CoV-2 Vaccination – Insights from a Self-Paid Tests at a General Internal Medicine Clinic

  1. We have also included our article: “Trends in Antibody Titers after SARS-CoV-2 Vaccination—Insights from Self-Paid Tests at a General Internal Medicine Clinic”, which is currently under submission, in the references.

L167-L168

“The medical records of a general internal medicine clinic in Wakayama, Japan, indicate that anti-receptor-binding domain (RBD) antibody titers are markedly elevated in breakthrough infections with COVID-19 after vaccination, as was found in this study [18].”

Please, highlight the strengths and weaknesses (or limitations) of your study.

  1. The strengths and limitations of the study have been newly added to the “Discussion” section.

L185-L195

“Although this data is based on a small number of studies at a single institution, using a measurement system that has already been commercialized and is widely used, we believe that it is a significant indicator of the changes in antibody titers over time after COVID-19 vaccinations, especially in cases combining breakthrough infections and COVID-19. In addition, some of these post-vaccination breakthrough infection cases were PCR-untested and had not been definitively diagnosed with COVID-19. This suggests that there may be hidden cases with very high antibody titers due to hybrid immunity among those who are asymptomatic, very mildly infected with COVID-19, unaware and undiagnosed. Although the existence of such cases is within the range of expectation, it is significant that we were able to use a real case in demonstration.

This study has several limitations. First, the size of the sample is small. Second, there is no information on SARS-CoV-2 variants. Third, the natural immunity to SARS-CoV-2 after vaccination has not yet been studied. In addition to the induction of antibodies, the effect of natural immunity is considered significant regarding hybrid immunity. As for cellular immunity after COVID-19 infection, the effects of CD4+ T cells and CD8+ T cells have been shown to be maintained for more than eight months [24,25]. The analysis of peripheral blood mononuclear cells (PBMCs) of COVID-19 patients is known to show impaired expression of inflammatory cytokines [26]. Although PBMCs were not analyzed in this study, future analysis of PBMCs from post-vaccination breakthrough infection cases may shed more light on the importance of natural immunity in "hybrid immunity" in COVID-19.”

ABSTRACT:

Line: 20-22 From June 2021 to February 2023, IgG-type SARS-CoV-2 antibody titers were measured in 127 participants, including 74 outpatients and 53 members of staffs, at the Osaka Dental University Hospital (65 males and 62 females, mean age 52.3 ± 19.1 years).

Line: 23-24 Consistent with previous reports, the SARS-CoV-2 antibody titer decreased with time, not only after the second dose, but also after the third dose of the vaccine if there is was no spontaneous COVID-19 infection.

Line: 185-187 In addition, the anti-receptor-binding domain (RBD) antibody, an antibody against SARS-CoV-2 186 spike protein, is widely used in Japan because it correlates with the SARS-CoV-2 neutralizing antibody titer (6,7) and can be measured at general medical facilities.

Line: 193-194 Anti-receptor-binding domain (RBD) antibody titers are known to increase markedly after the booster dose 193 (third dose) of the vaccine (10).

Line: 217-2020 SARS-CoV-2 antibody titers vary greatly from person to person and increase prominently considerably after spontaneous infection. In Japan, it is thought that many people have antibody titers of tens of thousands of AU/mL after spontaneous COVID-19 infection (often very mild and asymptomatic) following vaccination.

Line: 221-222 In Japan, a large number of patients are were infected with the Omicron strain in 2022 after receiving two doses of the COVID-19 vaccine in 2021.

Line: 222-225 As observed in this study, such cases are likely to maintain high antibody titers for more than six months after infection; however, in reality, even those with sufficiently high antibody titers are likely to have received a third or subsequent additional vaccination in a very large number of cases.

  1. We have attended to your note and accordingly corrected the relevant points in the text.

Reviewer 3 Report

dear Authors,

The present manuscript is an exciting study of a trending topic. I am finding of great enjoyable this analysis of how titering is maintained through time and the effect of infection over such titering.

The number of participants is not much but enough for confirming what others wider studies have done. The results are driving in the same direction regarding the efficacy and duration of the Covid vaccines.

On the other hand, I am missing a flow chart followed by the study and some demographic information about the participants.

The interest of this work is not in its novelty but in the conclusions obtained via a simple study.

Author Response

Reviewer 3

dear Authors,

The present manuscript is an exciting study of a trending topic. I am finding of great enjoyable this analysis of how titering is maintained through time and the effect of infection over such titering.

The number of participants is not much but enough for confirming what others wider studies have done. The results are driving in the same direction regarding the efficacy and duration of the Covid vaccines.

On the other hand, I am missing a flow chart followed by the study and some demographic information about the participants.

The interest of this work is not in its novelty but in the conclusions obtained via a simple study.

A. Thank you for your favorable comments. Please refer to Table 1, Table 2, and Figure 1 for a flowchart and demographic information.
